# Conserving the Diversity of Ecological Interactions: The Role of Two Threatened Macaw Species as Legitimate Dispersers of "Megafaunal" Fruits

**José L. Tella [1],\*, Fernando Hiraldo [1], Erica Pacífico [1], José A. Díaz-Luque [2], Francisco V. Dénes [1] , Fernanda M. Fontoura [3,4], Neiva Guedes [3,4] and Guillermo Blanco [5]**

[1] Department of Conservation Biology, Estación Biológica de Doñana, CSIC. Américo Vespucio s/n, E-41092 Sevilla, Spain; hiraldo@ebd.csic.es (F.H.); ericapacifico@ebd.csic.es (E.P.); voeroesd@ualberta.ca (F.V.D.)

[2] Foundation for the Research and Conservation of Bolivian Parrots (CLB). Estación Argentina, C/ Fermín Rivero 3460, Santa Cruz de la Sierra, Bolivia; saveparrot@gmail.com

[3] Universidade Anhanguera - Uniderp, Programa de Pós Graduação em Meio Ambiente e Desenvolvimento Regional, Rua Alexandre Herculano, 1400, Bairro Jardim Veraneio, 79037- 280 Campo Grande, Brazil; ferpa701@gmail.com (F.M.F); contato@institutoararaazul.org.br (N.G.)

[4] Instituto Arara Azul, Pesquisa em Conservação, Rua Klaus Sthurk, 106, Jardim Mansur, 79051-660 Campo Grande, Brazil

[5] Department of Evolutionary Ecology, Museo Nacional de Ciencias Naturales, CSIC. José Gutiérrez Abascal 2, 28006 Madrid, Spain; g.blanco@csic.es

\* Correspondence: tella@ebd.csic.es

**Abstract:** The extinction of ecological functions is increasingly considered a major component of biodiversity loss, given its pervasive effects on ecosystems, and it may precede the disappearance of the species engaged. Dispersal of many large-fruited (>4 cm diameter) plants is thought to have been handicapped after the extinction of megafauna in the Late Pleistocene and the recent defaunation of large mammals. We recorded the seed dispersal behavior of two macaws (*Anodorhynchus hyacinthinus* and *Anodorhynchus leari*) in three Neotropical biomes, totaling >1700 dispersal events from 18 plant species, 98% corresponding to six large-fruited palm species. Dispersal rates varied among palm species (5%–100%). Fruits were moved to perches at varying distances (means: 17–450 m, maximum 1620 m). Macaws also moved nuts after regurgitation by livestock, in an unusual case of tertiary dispersal, to distant perches. A high proportion (11%–75%) of dispersed nuts was found undamaged under perches, and palm recruitment was confirmed under 6%–73% of the perches. Our results showed that these macaws were legitimate, long-distance dispersers, and challenge the prevailing view that dispersal of large-fruited plants was compromised after megafauna extinction. The large range contraction of these threatened macaws, however, meant that these mutualistic interactions are functionally extinct over large areas at a continental scale.

**Keywords:** Caatinga; Cerrado; ecosystem services; Hyacinth macaw; Lear's macaw; megafauna; palms; Pantanal; plant-animal mutualisms; seed dispersal

## 1. Introduction

We are facing an era, the Anthropocene, characterized by unprecedented rates of human-driven biodiversity loss [1]. According to IUCN, 10%–30% of the amphibian, bird, and mammal species of the world are threatened by extinction [2], and extinction rates could be at least five times higher in the near future than in the recent past if current threats persist [3]. Added to the need to halt the decline and extinction of species, there is increasing concern on the loss of their ecological interactions [4,5].

Since all species are inter-connected through ecological interactions, the decline and extirpation of local populations, even in species not threatened with extinction, may lead to the disruption and loss of many interactions with far-reaching consequences for ecosystems [3–5]. As a response to the need for integrating the conservation of species with ecosystem conservation, a new framework has been recently proposed for incorporating the species' ecological functions into recovery planning [6].

Seed dispersal is a key ecological function played by vertebrate frugivores [7]. By influencing the demography of plants, seed dispersers shape the composition and abundance of plant communities and thereby the structure and functioning of ecosystems [8]. Anthropogenic impacts are eroding functional diversity of ecosystems by disrupting the interactions between frugivores and their food plants [9,10]. In particular, the decimation and extinction of large vertebrates are detrimental because of their important role as long-distance dispersers of large-seeded, biomass dominant, keystone plant species [9,11–13]. Currently, both large-seeded, long-lived plants, and their largest dispersers are seriously threatened by deforestation and defaunation to the extent that one or both partners, and their interaction, may become globally or functionally extinct in the near future [5,10,11]. This impact can be traced back to the possible human influence on the extinction of Pleistocene megafauna [11,14], i.e., mammals, such as giant sloths and gomphotheres, of seven genera with a body mass >1000 kg [15], which were thought to act in the past as dispersers of large fruits ("megafaunal fruits", usually >4 cm in diameter [15]) of many still extant plants. Although it has been argued that livestock partially filled the seed dispersal role of extinct megafauna [15–17], and of extant but largely decimated large mammals, such as tapirs [17,18], the presumably reduced dispersal of these large-fruited plants nowadays is widely recognized as a seed dispersal anachronism, the so-called megafaunal syndrome [15,16].

The evaluation of the influence of fruit and seed traits on dispersal mode by each fruit-eating organism is essential to understand plant demography and population dynamics [19]. Palms (Arecaceae) constitute a species-rich (>2500 sp) animal-dispersed plant family typical of tropical forests in which the potential role of extinct megafauna has recently received much attention. Onstein et al. [20] found that 12% of the palms of the world had large, megafaunal fruits and that species with small fruit sizes had increased speciation rates compared with those with megafaunal fruits. This result was suggested due to larger gene flow in palms with large-sized fruits due to the supposed ability of extinct megafauna to disperse them across large distances, compared to the more restricted dispersal of small fruits conducted by small-bodied frugivores [20]. Moreover, extinction rates of Neotropical palms with megafaunal fruits have increased since the onset of Quaternary, suggesting a concurrent role of climate oscillations, habitat fragmentation, and the loss of megafaunal dispersers [21]. Furthermore, a very recent synthesis of animal-mediated dispersal of palms suggests that the lack of a matching relationship between the size of fruits and frugivores in the Neotropics could be explained by the extinction of mammalian megafauna [22]. However, this study also emphasized the need for further research to address the large knowledge gap of palm-frugivore interactions [22], suggesting that the key dispersal role of some extant species, such as parrots, could have been overlooked [22,23].

Recent work has shown that several species of parrots, especially the largest macaw species, often disperse seeds in the Bolivian Amazonian savannas [24]. Strikingly, three macaw species of the genus *Ara* have been shown to be the main effective dispersers of the large-fruited (7–9 cm long, 4–5 cm diameter), biomass-dominant motacú palm (*Attalea princeps*), shaping the landscape structure of this biome and overriding the role of free-ranging livestock. These macaws are effectively contributing to forest regeneration and connectivity by dispersing palm fruits at high rates (75%–100% of the picked fruits) to distant (up to 1200 m) perching trees, where they consume the pulp and always discard the entire seeds [24]. However, this could be interpreted as a unique case since some macaw species, especially those of the genus *Anodorhynchus*, are pervasive seed predators [25]. The hyacinth macaw *Anodorhynchus hyacinthinus* (hereafter HM) is the largest parrot species in the world, while the congeneric Lear's macaw *A. leari* (hereafter LM) is phenotypically similar but somewhat smaller [26]. They have the strongest beaks among parrots, allowing them even to crack the nuts of large-sized palm fruits after defleshing them and discarding the mesocarp [25,27]. Therefore, these macaws might not

be contributing to legitimate seed dispersal despite having been observed transporting fruits in flight using their beaks or feet [17,23].

In this study, we tested the hypothesis that the two extant *Anodorhynchus* macaws might act as legitimate dispersers of large-fruited plants. This was prompted by recent findings showing that other parrot species, also considered pure seed predators, acted as seed dispersers within an antagonism-mutualism continuum [28]. As previously mentioned, *Anodorhynchus* macaws have been observed transporting fruits picked from the mother plants to distant perches for handling and consumption [17,23], but it is unknown whether all seeds are then predated, or a fraction of them survive predation and thus contributes to effective primary seed dispersal. On the other hand, these macaws have been reported actively searching for and predating on large palm seeds excreted by livestock, which has been interpreted as an evolutionary adaptation to exploit seeds excreted by extinct megafauna [27,29]. Similarly, it is unknown whether this tertiary dispersal process, following secondary dispersal by livestock, translates into effective seed dispersal. We relied on direct observations of foraging and fruit-dispersing macaws, complemented with camera trapping, as the most recommendable approach when the visual tracking of seed-dispersing vectors is affordable [30,31] (see also [24,32] for the same approach). Following this methodology, we were able to study all the subsequent steps of the multistage dispersal process: 1) fruit handling, 2) dispersal rates, 3) dispersal distances, 4) survival, and 5) germination of dispersed seeds (i.e., realized dispersal). Our results showed that *Anodorhynchus* macaws were frequent, long-distance, and effective dispersers of several plant species, mostly large-fruited palms. We thus discussed the key role of these two threatened macaw species [33,34] in the effective dispersal of large seeds, a role thus far mainly attributed to the extinct Pleistocene megafauna. Our findings highlighted how macaw population declines and range contractions might have further compromised the dispersal of large-fruited palms, and the need for recovery plans not only for their conservation but also to restore their ecological functions in the threatened ecosystems they inhabit.

## 2. Materials and Methods

### 2.1. Study Areas

Fieldwork consisted of 12 expeditions conducted in three biomes (Caatinga, Cerrado, and Pantanal) inhabited by LM and HM in Brazil and Bolivia. Table 1 shows the location of study areas, the macaw and palm species present in each, and fieldwork dates. The Caatinga biome, composed mostly of shrub and deciduous trees, constitutes the largest tropical dry forest region in South America and has remarkable rainfall variability from year to year [35,36]. The Cerrado biome is characterized by a heterogeneous savanna landscape, including grasslands, shrublands, gallery forests, and dry forests [37,38]. The Pantanal is the world's largest tropical wetland, with roughly 80% of its floodplains submerged during the rainy season [38]. Figure 1 shows the distribution of the three biomes and the two macaw species, as well as the location of the four study areas.

**Table 1.** Location of study areas, fieldwork dates, and the macaw and palm species present therein.

| Biome | Study Areas | Coordinates | Country | Fieldwork Dates | Macaw Species | Palm Species |
|---|---|---|---|---|---|---|
| Caatinga | Raso da Catarina | 10° 17.089′ S, 38° 42.419′ W | Brazil | August, September 2014; February, | *A. leari* | *Syagrus coronata* |
| | | | | April, May 2015, March-May 2016 | | |
| Cerrado | São Gonçalo | 10° 06.023′ S, 45° 22.228′ W | Brazil | June 2015, October 2016, | *A. hyacinthinus* | *Attalea barreirensis* |
| | da Gurguéia | | | January 2017 | | *Attalea eichleri* |

**Table 1.** *Cont.*

| Biome | Study Areas | Coordinates | Country | Fieldwork Dates | Macaw Species | Palm Species |
|---|---|---|---|---|---|---|
| | | | | | | *Mauritia flexuosa* |
| Pantanal | Fazenda Caiman | 19° 57.263' S, 56°18.258' W | Brazil | November 2015 | *A. hyacinthinus* | *Acrocomia totai* |
| | | | | | | *Attalea phalerata* |
| Pantanal | San Matías | 17° 13.437' S, 58° 36.700' W | Bolivia | November 2017 | *A. hyacinthinus* | *Acrocomia totai* |
| | | | | | | *Attalea phalerata* |

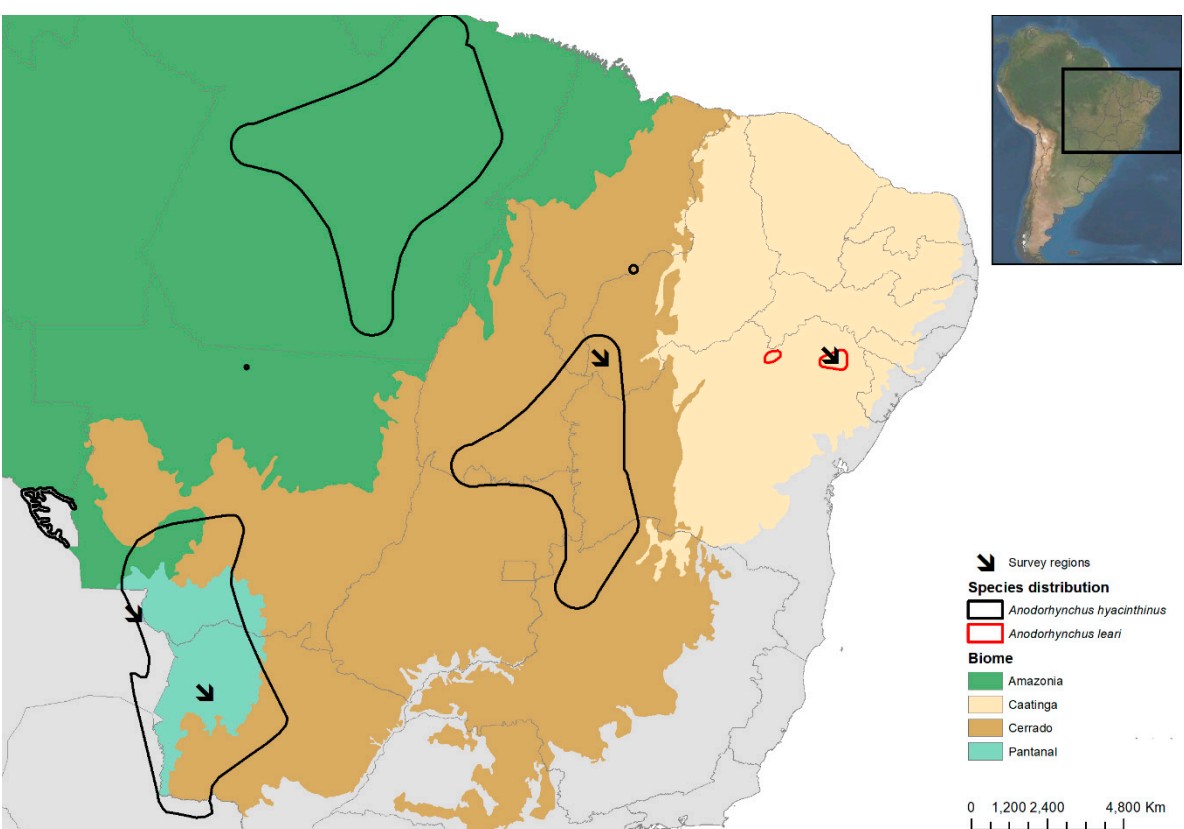

**Figure 1.** Global distribution of the two macaw species studied in Brazil and Bolivia, biomes, and location of the four study areas (arrows). Maps were generated using ArcGIS v.10.5. (ESRI, Redlands, USA), using biome layers [39], Landsat-4 images [40], and species distributions available [33,34].

*2.2. Plant Species Dispersed*

We identified the plant species dispersed by macaws using a variety of plant guides. We found only one case of conflicting identification: we identified a palm species typical of the Pantanal biome, previously considered as *Acrocomia aculeata*, as *Acrocomia totai* based on the distribution range of both species [41]. The size of fruits and seeds and the number of seeds per fruit of the plant species dispersed were obtained from the literature [15,41–44] and our measurements. We classified type I megafauna-dependent plants as those with fruits with a mean diameter >4 cm and usually containing up to five seeds [15]. This is a widely accepted criterion used to identify palms with megafaunal fruits [20,21]. However, the same authors included some smaller fruits (2–3 cm diameter) in their list of megafauna-dependent species [15]. This inconsistency in the definition led to uncertainty in the classification of some of our studied species (see Results).

### 2.3. Foraging and Dispersal Behavior and Dispersal Rates

We traveled through the foraging areas of the species and stopped when we found a group of foraging macaws. We then observed their foraging behavior with binoculars and telescopes from a distance to avoid disturbance (see [28,32,45] for the same methodology). We recorded how macaws handled fruits (consuming or not the seeds) and any event in which they dispersed fruits from the mother plant (primary dispersal) or seeds excreted by livestock (tertiary dispersal).

The proportion of fruits dispersed (dispersal rates) was obtained in detail only from four palm species. Dispersal rates of two palm species (*Acrocomia totai* and *Attalea phalerata*) by HM were obtained through direct observations in the Bolivian Pantanal. Similar to other parrot species, macaws often move within and among canopies, making it difficult to focus on individual birds. We, therefore, recorded the total number of fruits consumed on the mother palm and the total number of fruits dispersed to distant perches by any individual observed to calculate the flocks' dispersal rate (see [24,32,46] for the same methodology). On the other hand, the fruits of the two bush-layer palm species present in the Cerrado (*Attalea barreirensis* and *Attalea eichleri*) are almost at ground level, thus making direct observations of foraging macaws from a distance difficult. To solve this problem, we used infrared-triggered camera traps to obtain fruit dispersal rates by HM. Cameras (20 for *A. barreirensis* and 15 for *A. eichleri*) were placed and hidden at ground level close (3–5 m) to the palms and ran automatically for five consecutive days. Cameras were motion-activated, obtaining multiple instantaneous digital captures (every 5 s), and thus they allowed to record complete sequences of macaws removing and dispersing the fruits. Distant palms with mature fruits not preyed upon were selected randomly. One camera placed on an *A. barreirensis* failed due to rain, and thus the information was obtained from 34 fruiting palms for a total of 4080 h.

### 2.4. Dispersal Distances

When dispersal events were observed, we identified the plant species and measured the distance from the point where the fruit was picked to the point where the bird perched for fruit handling. This was measured with a laser rangefinder incorporated into binoculars (Leica Geovid 10 × 42, range of measurements: 10–1300 m). When flying dispersers were lost from sight in the vegetation, we conservatively considered a minimum dispersal distance from the location where the fruit was collected to the location where the macaw was no longer visible. For dispersers already flying when they were first observed, we recorded the distance at which they were first detected and then followed them with binoculars. In these cases, we also considered minimum dispersal distances, as measured from the location of the first sighting to where the disperser perched for handling the fruit, where they released the fruit in flight, or where they were lost from sight in flight. In other instances, we found seeds under perches (trees, cliffs, and fence poles), where we observed macaws consuming them, and conservatively estimated the minimum dispersal distance as the distance to the closest fruiting female plant (see [17,32,46] for the same methodology).

### 2.5. The Proportion of Surviving Dispersed Seeds and Realized Dispersal

This information was only obtained for the fruits dispersed from palms. After locating perching sites where we observed macaws handling and consuming the dispersed palm fruits, we carefully searched under the perches for both predated and undamaged seeds to calculate the proportion of seeds surviving predation by macaws. These proportions could both underestimate or overestimate the actual proportion of seeds surviving predation by macaws since we could not discard that mammals could remove some undamaged seeds before or after our sampling. Therefore, the proportions obtained should be interpreted with caution. Additionally, we searched for germinating seeds and young saplings (i.e., plants <50 cm high) to verify realized dispersal [30]. In most cases, the perching tree—dead or alive—was a species different from the seed species found below, while, in a few cases, perching sites were non-fruiting trees of the same species. We did not observe other species using these

perching sites that could disperse the large-sized palm fruits. Therefore, we could assume that the seeds we found under perches were transported due to the dispersal behavior of macaws (see [24,32] for the same reasoning).

## 2.6. Statistical Analyses

We analyzed differences in dispersal rates and in the proportion of undamaged dispersed seeds among palm species using contingency tables and Chi-square tests. We used Spearman correlation to test the potential relationship between fruit sizes (estimated as the product of diameter by length) and estimated mean dispersal distances.

Dispersal distance distributions were right-censored since they included a high proportion of right-censored dispersal distances (e.g., we recorded many flying birds carrying fruit in the bill until they were out of sight, and thus measured the distance to the last point of observation, henceforth identified as minimum dispersal distance). We thus relied on failure-time analysis for estimating actual mean dispersal distances (see [32,46] for the same approach). Briefly, we employed an adaptation of Kaplan–Meier (or Product-Limit) estimators for survival functions [47] to estimate dispersal functions, D(d), which inform the probability that a dispersal event would occur at a given distance. The Kaplan–Meier estimator provides an efficient means of estimating the dispersal function for right-censored data, such as our dispersal dataset [32,46], in which both observed (exact distances) and unobserved (minimum distances) events were recorded. The Kaplan–Meier estimate of D(d) corresponds to the non-parametric maximum likelihood estimation of D(d) and is a step function with jumps at the observed event distances. The size of these jumps depends not only on the number of events observed at each event distance $d_i$ but also on the pattern of the censored observations before $d_i$. We estimated D(d) for the two species of macaws and for different plant species separately and tested for significant differences in dispersal distances for each group based on a generalized Wilcoxon test [47,48]. Mean and median dispersal distances were obtained from the estimated functions. The mean is the integral of the dispersal curve, conservatively restricting the mean to an upper limit that corresponds to the larger minimum distance recorded for each species. The median is the intersection of the curve with a horizontal line drawn at 0.542. We used the package survival [49] in R [50] 43 to estimate dispersal functions and perform significance tests.

## 2.7. Ethics Statement

This study relied on observational data obtained in areas unrestricted to people and thus did not require special permits except for the San Matías study area (permits MMAYA/VMABCCGDF/DGBAP/MEG N° 0151/2017 from Dirección General de Biodiversidad, MMyMA, Bolivia) and Raso da Catarina (Researcher's Licenses SISBIO 12763-7, 2991/5, Brazil). This study did not require ethical approval by our research institutions as it did not involve experimental work or invasive methods with animals.

## 3. Results

### 3.1. Plants Dispersed

We recorded a total of 1722 dispersal events by the two macaw species (1590 through direct observations and 132 through camera trapping) in the three biomes surveyed. The fruits were from 18 plant species from eight families (Table 2). Six species were clearly classified as type I megafaunal fruits (Table 2, see [15] for criteria). However, the same authors proposing these criteria also included *Spondias tuberosa* and five palm species with smaller fruit sizes (2.2–3.5 cm diameter), as megafauna-dependent plants [15]. Therefore, the classification of three of our study species (*Spondias tuberosa*, *Syagrus coronata*, and *Acrocomia totai*) remained uncertain. Most of the fruits dispersed (97.7%) were from six palm species, and most, if not all of them, could be considered as megafauna-dependent plants (Table 2).

**Table 2.** Plant species dispersed by hyacinth and Lear's macaws in the three studied biomes, size of their fruits and seeds (in mm), number of seeds by fruit, and number of primary (PD) and tertiary (TD) dispersal events with the range of dispersal distances observed (in m). Fruits considered as megafauna-dependent type I fruits (MF, following [15]) are indicated (see results for uncertainty for some species).

| Plant species | Family | Biome | Fruit Size | Seed Size | N Seeds | Source | MF | PD | TD | Distance Range |
|---|---|---|---|---|---|---|---|---|---|---|
| **Lear's macaw** | | | | | | | | | | |
| *Anacardium occidentale* | Anacardiaceae | Caatinga | 90 × 40 | 40 × 25 | 1 | [15] | Yes | 3 | | 160–600 |
| *Cereus jamacaru* | Cactaceae | Caatinga | 82.3 × 62.6 | 2.62 × 1.73 | 1400 | own | | 2 | | 50–127 |
| *Colicodendron (Capparis) yco* | Capparaceae | Caatinga | 67.6 × 42.4 | 42.4 × 12.9 | 17 | own | Yes | 2 | | 10–10 |
| *Dioclea grandiflora* | Fabacedae | Caatinga | 115 × 40 | 25 × 25 | 2–5 | [42] | | 1 | | 150 |
| *Jatropha sp.* | Euphorbiaceae | Caatinga | | | | | | 4 | | 119–348 |
| *Jatropha mollisima* | Euphorbiaceae | Caatinga | 19.3 × 18.7 | 9 × 6 | 3 | own | | 10 | | 4–25 |
| *Manihot sp.* | Euphorbiaceae | Caatinga | | | | | | 1 | | 4 |
| *Pilosocereus pachycladus* | Cactaceae | Caatinga | 50.5 × 38.1 | 1.89 × 1.35 | 3800 | own | | 7 | | 6–1000 |
| *Spondias tuberosa* | Anacardiaceae | Caatinga | 45 × 38 | 28 × 20 | 1 | [15]; own | Yes? | 1 | 2 | 12–432 |
| *Syagrus coronata* | Arecaceae | Caatinga | 28.8 × 24.3 | 22.3 × 15.5 | 1 | own | Yes? | 362 | | 3–250 |
| *Zea mays* | Poaceae | Caatinga | 167.6 × 47.4 | 1.09 × 0.93 | >100 | own | | 2 | | 33–1000 |
| **Hyacinth macaw** | | | | | | | | | | |
| *Acrocomia totai* | Arecaceae | Pantanal | 27.8 × 27.8 | 16.1 × 16.1 | 1 | [41]; own | Yes? | 300 | 114 | 1–400 |
| *Attalea barreirensis* | Arecaceae | Cerrado | 59.1 × 40.7 | 32 × 11.5 | 1–4 | own | Yes | 352 | | 3–1620 |
| *Attalea eichleri* | Arecaceae | Cerrado | 60.5 × 40.5 | 37.5 × 11.5 | 1–5 | [41,43] | Yes | 409 | | 1–223 |
| *Attalea phalerata* | Arecaceae | Pantanal | 57.5 × 40 | 45 × 25 | 2–4 | [41,43] | Yes | 132 | 3 | 4–1011 |
| *Mauritia flexuosa* | Arecaceae | Cerrado | 57.5 × 57.5 | 35 × 25 | 1 | [15] | Yes | 11 | | 40–234 |
| *Spondias mombin* | Anacardiaceae | Pantanal | 30 × 22.5 | 25.5 × 15.5 | 4–5 | own | | 1 | 2 | 1–210 |
| *Vitex cymosa* | Lamiaceae | Pantanal | 29 × 25 | 17 × 10 | 1 | [44] | | 1 | | 40 |

### 3.2. Fruit Handling and Dispersal Behavior

The two macaw species defleshed palm fruits just after picking them from the mother plant (Figure 2a) to later break the nut and consume the seeds. The only exception was the palm *Mauritia flexuosa*, from which fruit HM only consumed the pericarp, discarding the whole nut with undamaged seeds. However, macaws often transported fruits to distant perching sites (Figure 2b,c). Palm fruits were mostly individually dispersed, although, in some instances, HM carried two fruits of *A. barreirensis* and *A. eichleri* (Figure 3b), and LM carried fragments of infructescences containing several fruits of *S. coronata* (Figure 2e). Fruits were mostly dispersed by carrying in the bill but were sometimes carried with the feet (Figure 2d). In most cases (92.6%), observations corresponded to primary seed dispersal, i.e., the fruit was picked from the mother plant. Moreover, we also recorded 121 instances of tertiary seed dispersal (i.e., after regurgitation by cattle and goats, Figure 2g,h) by HM and LM, most of them (96.7%) corresponding to two palm species (Table 2).

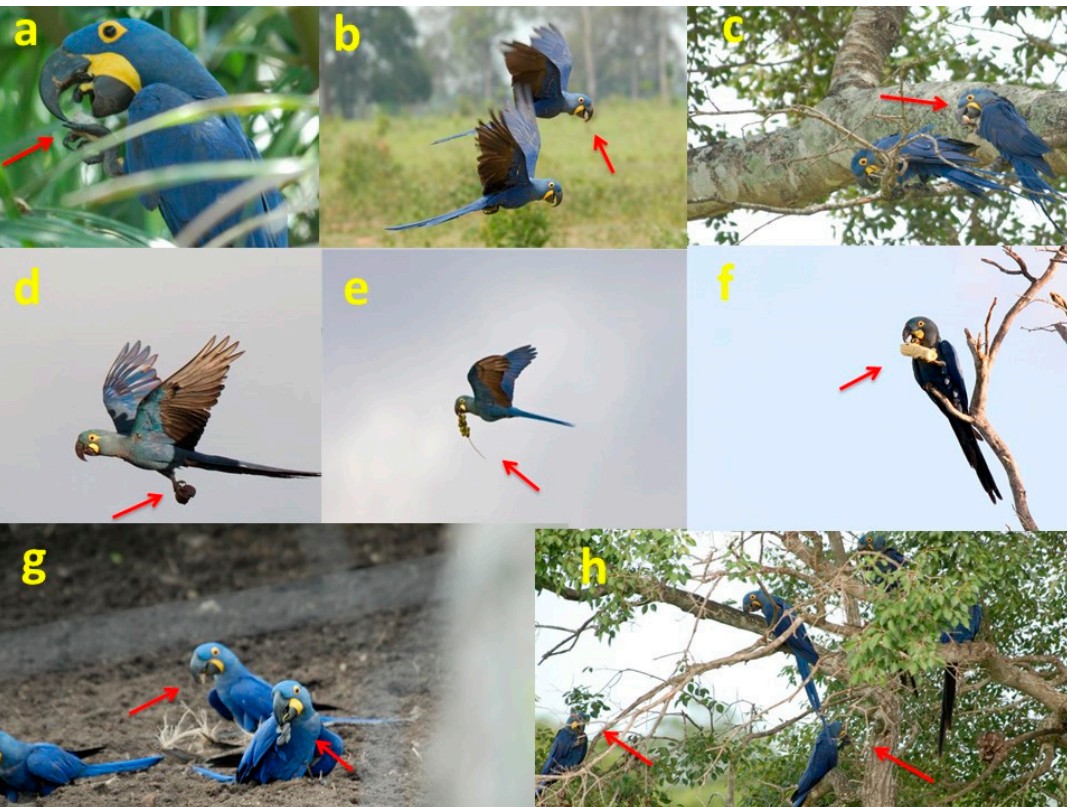

**Figure 2.** Seed dispersal processes. (**a**) Hyacinth macaw defleshing an *Attalea phalerata* palm fruit; the macaw dropped the fully defleshed but undamaged nut under the same mother palm tree. Primary dispersal: (**b**) one hyacinth macaw flew a minimum distance of 160 m carrying a defleshed *A. phalerata* fruit to be handled, (**c**) in a perching tree. Large cactus fruits are carried by foot (**d**), while a bunch of fruits of *Syagrus coronata* palm is carried by beak (**e**) by Lear's macaws. (**f**) A corn cob dispersed by Lear's macaw is handled in a perching tree. Tertiary dispersal: (**g**) Hyacinth macaws picked *A. phalerata* and *Acrocomia totai* nuts from the ground after regurgitation by cattle and flew 80 m to a perching tree (**h**) to handle them and consume the seeds. Some undamaged nuts were dropped under perches. Photographs taken by J.L. Tella (a,b,c,g,h), Steve Brookes (d), and Ciro Albano (e,f).

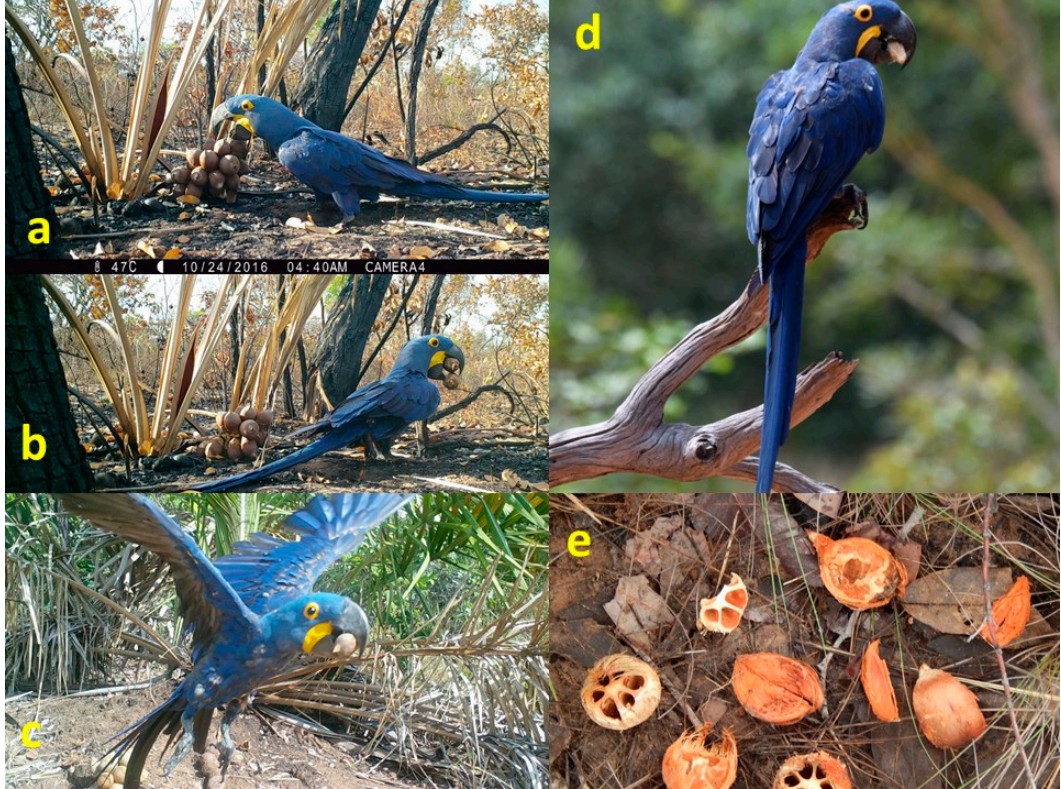

**Figure 3.** Dispersal process of bush layer palms (*Attalea barreirensis* and *Attalea eichleri*), as revealed by camera trapping. Hyacinth macaws landed and walked to the ground-level bunches of fruits (**a**) to pick up one or two at once and walked to the surroundings (**b**) to deflesh and ultimately consume the seeds, or they flew (**c**) to distant perching trees (**d**) to handle the fruits. Defleshed, predated, and undamaged nuts were easily found under perches (**e**). Photographs were taken by camera traps (a,b,c), Manuel de la Riva (d), and Fernando Hiraldo (e).

### 3.3. Primary Dispersal Rates

We observed the feeding behavior of 23 flocks of HM on the fruits of *A. totai*. Overall, they dispersed 4.75% of the fruits picked from the mother palm ($n = 316$) to distant perching sites. Only one flock of HM was observed feeding on *A. phalerata* fruits, which dispersed 13.3% of the fruits handled ($n = 15$).

Camera traps allowed us to estimate dispersal rates for the two bush-layer palm species (*A. barreirensis* and *A. eichleri*). HM visited four of the 19 (i.e., 21%) monitored *A. barreirensis*, thus rendering a visiting rate of 4.2% of the available palms per day (i.e., 4 visits/19 palms x 5 monitoring days). Between one and four macaws were simultaneously recorded by camera traps removing fruits (Figure 3a) on consecutive visits to the same palm within the same day. Overall, 71.5% of the fruits available on the visited palms ($n = 151$) were removed. In all cases, macaws walked or flew outside of the camera's range (Figure 3b,c), transporting the fruits to a perching tree (Figure 3d), where they handled them for consumption. Therefore, 100% of the fruits removed from the mother palm were dispersed.

Camera traps placed close to *A. eichleri* palms registered visits by HM in two out of 15 monitored palms (i.e., 13.3%), involving, in both cases, at least three different individuals visiting the palm on the same day. Therefore, 2.7% of the available (monitored) palms were visited per day, and 61.5% of the fruits available on the visited palms ($n = 39$) were removed by macaws, showing the same behavior as when removing *A. barreirensis* fruits (i.e., 100% of the removed fruits were dispersed).

Dispersal rates were much higher in the two bush-layer (100%) than in the two tree-shaped palm species (4.75%–13.3%; $X^2 = 387.3$, df $= 1$, $p < 0.0001$).

### 3.4. Dispersal Distances

Primary dispersal distances ranged from 1 m to a minimum of 1620 m in HM (*n* = 1074, Figure 4a) and from 3 m to a minimum of 1000 m in LM (*n* = 397, Figure 4b). Overall, 81.9% of the records corresponded to minimum dispersal distances. Figure 4c shows dispersal probabilities estimated, taking into account the censored (minimum) dispersal distances recorded. In the case of HM, only 32.2% of the recorded distances were censored. The estimated mean dispersal distance for this species was 195 m (SE = 32.4), with an estimated median of 40 m (95% CI: 39–70 m). LM showed a much larger estimated mean dispersal distance (874.5 m, SE = 97.5). A median dispersal distance could not be obtained for this species, given that most recorded distances (98%) were censored, and the dispersal curve did not cross the horizontal line at 0.5 (Figure 4c). Dispersal functions differed significantly between the two macaw species (generalized Wilcoxon test, $X^2$ = 270, df = 1, $p < 0.0001$; Figure 4c).

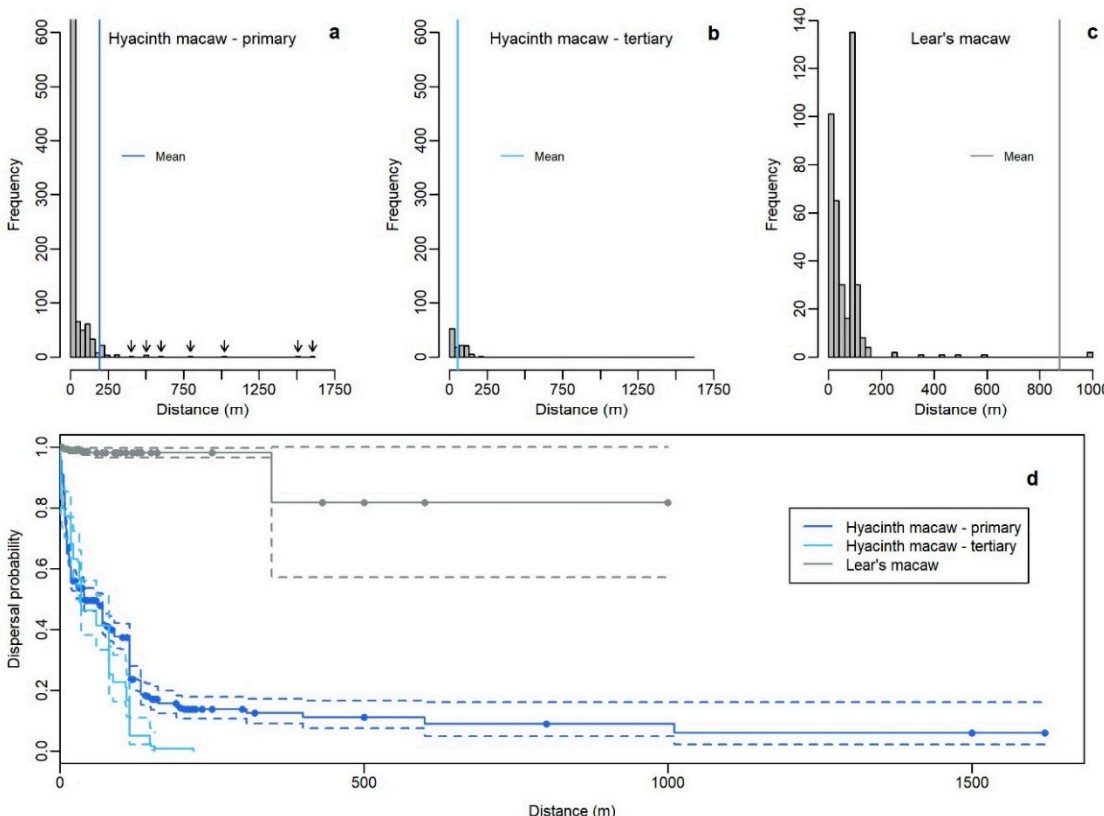

**Figure 4.** Distribution of fruit dispersal distances recorded for primary (**a**) and tertiary (**b**) dispersal in hyacinth and Lear's macaws (**c**). Vertical lines show the estimated mean dispersal distances. (**d**) Kaplan–Meier estimates of dispersal functions; dashed lines show 95% confidence bounds, and solid circles indicate censored (minimum distance) observations.

Dispersal distances significantly varied among the five palm species dispersed by HM ($X^2$ = 668, df = 4, $p < 0.0001$), with mean estimates ranging from ca. 17 m for *A. eichleri* to 454 m for *A. phalerata*, while distances for *S. coronata* dispersed by LM fell within this range (Table 3). However, mean dispersal distances were uncorrelated to fruit size in these six palm species (Spearman correlation, $r_s$ = –0.43, $p = 0.42$, $n = 6$, Figure 5).

**Table 3.** Estimated mean (+SE) and median (95% CI) dispersal distances for all primary and tertiary dispersal events and each palm species, indicating the total number of recorded dispersal distances (Nt), the number of recorded exact distances (Ne), and the upper limit (i.e., the maximum distance observed) used for calculating the mean dispersal distance for each species (UL).

|  | Plant | Nt | Ne | Mean | SE | Median | 95% CI | UL |
|---|---|---|---|---|---|---|---|---|
| Hyacinth macaw primary | all dispersions | 1074 | 553 | 195.3 | 32.4 | 40 | 39–70 | 1620 |
|  | *Acrocomia totai* | 300 | 28 | 162.4 | 19.7 | 150 | 85–161 | 400 |
|  | *Attalea barreirensis* | 244 | 116 | 218.9 | 46.2 | 115 | 115–115 | 1620 |
|  | *Attalea eichleri* | 385 | 368 | 17.1 | 1.6 | 9 | 9–10 | 223 |
|  | *Attalea phalerata* | 132 | 40 | 453.6 | 59.4 | 90 | 70–NA | 1011 |
|  | *Mauritia flexuosa* | 11 | 1 | 216.4 | 16.8 | NA | NA–NA | 234 |
| Hyacinth macaw tertiary | all dispersions | 119 | 119 | 55.2 | 4.33 | 35 | 32–80 | 220 |
|  | *Acrocomia totai* | 114 | 114 | 56.5 | 4.4 | 35 | 32–80 | 220 |
| Lear's macaw primary | all dispersions | 397 | 7 | 874.5 | 97.5 | NA | NA–NA | 1000 |
|  | *Syagrus coronata* | 362 | 2 | 248.6 | 10.1 | NA | NA–NA | 250 |

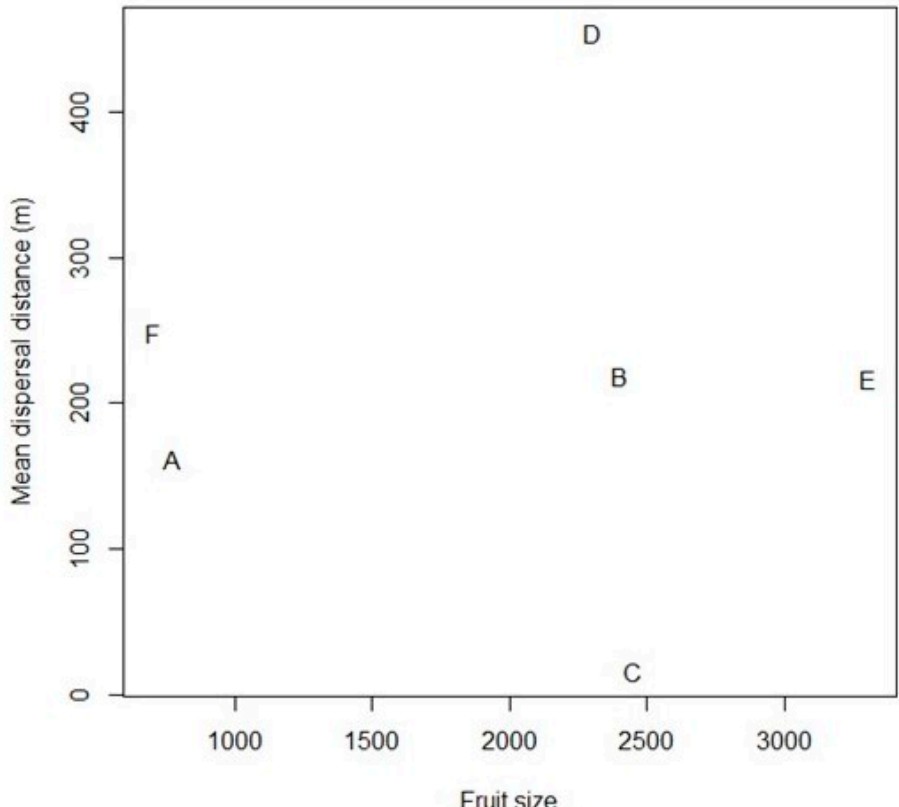

**Figure 5.** Relationship between estimated mean dispersal distances and fruit sizes (product of length and diameter in mm) in the six studied palm species (A: *A. totai*, B: *A. barreirensis*, C: *A. eichleri*, D: *A. phalerata*, E: *M. flexuosa*, F: *S. coronata*).

Tertiary dispersal distances ranged from 1 to 220 m in HM ($n = 119$, all of them as exact distances), and most corresponded to nuts of *A. totai* dispersed after regurgitation by goats and cattle (Table 3). Mean tertiary dispersal distance of *A. totai* was about one-third (56.5 m, SE = 4.4, $n = 114$) of the mean primary dispersal distance for the same species (162.4 m, SE = 19.7, $n = 300$; $X^2 = 95.8$, df = 1, $p < 0.0001$; Table 3).

*3.5. The Proportion of Surviving Dispersed Seeds*

We found dispersed palm nuts containing undamaged seeds under 65.2% of the 181 inspected perching sites. Overall, 16.6% of the dispersed palm nuts (*n* = 1115) found under these perching sites were undamaged nuts (defleshed or not) with ripe seeds. There were no significant differences among palm species in the proportion of perching sites where we recorded undamaged nuts ($X^2$ = 7.06, df = 5, *p* = 0.216). However, the proportion of undamaged nuts found under perching sites varied among species ($X^2$ = 71.2, df = 5, *p* < 0.001), ranging from 11% (*A. eichleri*) to 75% (*M. flexuosa*, the only palm species from which macaws usually discard their seeds) (Table 4).

**Table 4.** Percentage of perching sites where macaws dispersed undamaged nuts, and the percentage of undamaged nuts found under these perching sites for each palm species.

| Species. | Biome | % Perching Sites | N | % Undamaged Nuts | N |
|---|---|---|---|---|---|
| Lear's macaw | | | | | |
| *Syagrus coronatus* | Caatinga | 57.4 | 47 | 13.4 | 231 |
| Hyacinth macaw | | | | | |
| *Acrocomia totai* | Pantanal | 30.8 | 13 | 18.6 | 220 |
| *Attalea barreirensis* | Cerrado | 57.7 | 26 | 15.6 | 224 |
| *Attalea eichleri* | Cerrado | 73.2 | 71 | 10.9 | 357 |
| *Attalea phalerata* | Pantanal | 55 | 20 | 44 | 75 |
| *Mauritia flexuosa* | Cerrado | 100 | 4 | 75 | 8 |

*3.6. Realized Seed Dispersal*

In Caatinga biome, we recorded 114 *S. coronata* saplings under 23.4% of the 47 perching sites used by LM for handling dispersed fruits, with 2–38 saplings per site (mean = 10.4). In the Cerrado biome, we could not differentiate between saplings of *A. barreirensis* and *A. eichleri* under the 90 perching sites used by HM. Saplings of these *Attalea* sp. palms were found under 73.3% of these perching trees, with 1–28 saplings per site (mean = 5). However, we only found five saplings of *A. phalerata* in two (6.1%) of the 33 perching trees inspected in the Pantanal biome. Although we could not collect information on seed viability, these results showed that a number of the dispersed seeds were viable and germinated.

## 4. Discussion

*4.1. Anodorhynchus Macaws as Frequent, Long-distance Dispersers of Large-fruited Plants*

It has been argued that no present-day Neotropical frugivore, with the probable exception of tapirs and introduced livestock, is likely to provide dispersal services, combining reliable consumption and removal of seeds >2.5 cm diameter on a regular basis [15]. Following this argument, the survival of megafauna-dependent plants since the Pleistocene could be explained by the action of less efficient secondary or sporadic primary dispersers, anthropogenic, and abiotic factors [15]. We did not aim to identify here the potential disperser coteries for several palm species [17], but rather we endeavored to test the hypothesis that two macaw species could serve as their legitimate, long-distance seed dispersers. This potential plant-animal mutualism has been so far highly unexpected since, contrary to other macaws that only consume the pulp and discard the seeds of palms [24], *Anodorhynchus* macaws are able to crack the woody coat to consume the seeds that dominate their diet. Thus, they have been exclusively considered as plant antagonists [25]. However, this role as apparently pure seed predators can ultimately represent a mutualistic (or conditional) relationship [28] if a functionally relevant proportion of the seeds is successfully recruited from macaw dispersal events, as we have demonstrated. This may be especially true for long-lived plant species with a reproductive strategy, maximizing the number of seeds relative to the resources invested in them to cope with scarce recruitment [51]. These species have been argued to primarily invest in the simultaneous production of

large crops of large seeds, swamping and satiating predators, that also act as seed dispersers [46,52,53]. Moreover, the effective dispersal of a small proportion of seeds may play a key role in the demography of large-fruited plants [15].

Parrots are able to exploit multiple foraging opportunities, encompassing virtually all plant parts in all maturation stages of a widesr variety of species [54]. However, *Anodorhynchus* macaws can be considered palm specialists [26–29], and, as we expected, they play a key role as dispersers of mostly large-fruited palms. Our results contributed to filling a large knowledge gap of palm-frugivore interactions, recently highlighted by Muñoz et al. [22], and supported their suggestion that parrots might play a role as palm seed dispersers. Our combination of camera trapping with direct observations showed dispersal rates even higher than those recorded for other parrots and macaws [24,46,55]. Dispersal rates varied, however, among palm species, with the highest (100%) for the two bush-layer palms. The decision to move with food and the distances traveled can depend on plant and fruit traits (e.g., size, structure) and on parrots' morphological traits and other factors, such as the abundance and accessibility of food, the density of competitors, presence of predators, etc. [28]. Macaws may move palm fruits to distant perches simply for a more comfortable site to handle them for consumption [24]. In the case of fruits from ground-level bush layer palms, they may move them more frequently as a cautionary behavior against terrestrial predators, such as felids and large-sized snakes, that occasionally prey on them (L. Lima pers. com.). LM move palm fruits but are also long-distance seed dispersers of large-fruited plant species with tiny seeds (cacti) and, unlike other frugivores, they act both as endozoochorous and ectozoochorous dispersers of these species [56].

In addition to primary dispersal, the consumption by HM of seeds secondarily dispersed by livestock (and presumably by megafauna in the past) has previously been observed without reporting subsequent seed dispersal [27]. Here, we confirmed the dispersal of seeds of four plant species (mostly palms) by HM and LM after regurgitation by cattle and goats. Fruits passively dropped from palms can be consumed and dispersed by livestock [17,24], and the generally unknown distance of these secondary dispersal events can increase and change in direction after tertiary dispersal by macaws. Importantly, we found that macaws moved regurgitated palm nuts across relatively large distances (up to 220 m) from sites, where livestock concentrate to rest and the regurgitated nuts accumulate in high numbers. The likelihood of plant recruitment is low in these sites due to trampling and soil compaction by livestock [24]. Therefore, tertiary dispersal by macaws may make these ineffective secondary dispersal events effective, thus benefiting the plant. On the other hand, macaws may benefit from the pulp-cleaning resulting from the livestock gut passage [27]. A similar process has been described involving cassowaries (*Casuarius* spp.) as endozoochorous dispersers of the Ti tree (*Terminalia impediens*) fruits, whose pulp (removed in the cassowary gut) can act as a mechanical deterrent for predation by palm cockatoos (*Probosciger aterrimus*) on the tree [57]. However, the subsequent potential cockatoo seed predation and dispersal from the cassowaries' feces remains unclear [25,57].

In this study, the observed dispersal distances ranged from a few meters to >1600 m, and the estimated mean distances varied between the two macaw species and among the palm species dispersed. Variation in dispersal distances among palm species was unrelated to their fruit sizes, suggesting that fruit size does not constrain dispersal by macaws and that such variability may rather result from variation in the spatial distribution of adequate perching sites within and among biomes. Differences in the spatial distribution of perches could also explain the differences between primary and tertiary dispersal distances of *A. totai* by HM. It is worth noting, however, that our mean dispersal distance estimates were surely underestimated even though we accounted for right-censored data in the analyses [32,46] (see also [17] for much shorter distance estimates when not accounting for right-censored data). While short dispersal distances were often measured exactly, the largest distances recorded and used as the upper restriction limit for analyses corresponded to minimum distances. That is, we observed fruit-carrying macaws coming from long distances when they were first detected, and they continued out of sight while still transporting them in flight. Thus, the actual distances could be a number of km longer, leading to an underestimation of maximum and

mean dispersal distances. Nevertheless, most of our estimated mean and median primary dispersal distances (Table 3) were well above 100 m—a distance threshold often used to define long-distance seed dispersal [58]. These distances were shorter but within the range of those estimated through simulations for the Pleistocene megafauna, suggesting that extinct large-bodied mammals would frequently disperse large seeds over a thousand meters, whereas smaller-bodied species were more likely to deposit the seeds over a few hundred meters [59]. Nonetheless, macaws disperse seeds at farther distances than a scatter-hoarding rodent (with maximum dispersal distances <400 m), which has been recognized as a long-distance disperser of a megafaunal palm fruit [58]. In any case, the combination of short and very large dispersal distances seems to be the rule in the mutualistic interaction of macaws and other parrots with their food plants [23,24,32,46], and it can determine the spatial distribution, genetic structure, and population dynamics of these species [24]. In particular, despite being underrepresented here, long-distance seed dispersal can be especially relevant for genetic interchange in plant populations [60–63], although focused studies are required to assess the impact of *Anodorhynchus* macaws on these processes.

## 4.2. Rates and Locations of Effective Seed Dispersal and Recruitment

Knowledge of the dispersal location of seeds moved by particular organisms is scarce due to the logistic challenges involved [30,31,59,64]. As for other macaw and parrot species [24,32], the observation of fruit-carrying individuals flying to perching sites makes the estimation of effective dispersal rates easier. Pooling all palm species, about 17% of dispersed fruits found below perching sites were undamaged or only defleshed and thus contained undamaged seeds, as a result of the generalized food-wasting behavior of parrots [65]. Importantly, these presumably viable nuts were found in a high proportion (65%) of the perching sites. Therefore, despite the fact that *Anodorhynchus* macaws prey upon a high proportion of the handled nuts, they are contributing to their dispersal over the large areas covered daily to track fruiting plants throughout the year. Moreover, undamaged dispersed seeds of plants other than palms (*Jatropha mollisima, Anacardium occidentale, Colicodendron yco, Cereus jamacaru, Spondias mombim, Vitex cymosa*) were also found at lower frequencies under perching trees.

In spite of its potential influence on population dynamics, there is scarce detailed information on the outcome of parrot dispersal on seed germination and sapling recruitment [24,32,46]. We confirmed realized dispersal by finding a variable number of palm saplings below 23% and 73% of the macaw's perching sites in the Caatinga and Cerrado biomes, respectively. The scarcity of saplings under perching trees in Pantanal could be explained by a high density of livestock (authors, pers. obs.), which could reduce seedling recruitment through overgrazing and trampling [24,66,67]. In fact, the negative impacts of cattle on palm recruitment have been demonstrated in the Pantanal [68]. Overall, the presence of viable seeds and saplings below numerous perching sites at variable distances from mother plants shows that the overlooked long-distance dispersal exerted by *Anodorhynchus* macaws is effectively translated into a successful plant recruitment, and thus they can play an important role in ecosystem structure and functioning [24,28,45].

## 4.3. Palm-Macaw Evolutionary Relationships and the Megafauna Syndrome

The large dependence of *Anodorhynchus* macaws on palms, as well as their role as legitimate dispersers of their seeds, suggests intimate relationships with conditional antagonistic-mutualistic outcomes for both partners and potential co-evolution of traits [69]. Specifically, the antagonistic seed predation interaction could be driven by an arms-race involving the extremely hard coat of palm fruits and the escalating huge beaks of macaws to crack the nuts and consume the seeds [25,27]. The outcome of this evolutionary race could promote a release effect exploited by palms to exclude seed predators other than macaws, thus increasing the likelihood of long-distance dispersal by these mobile and strong flyers. Therefore, as a result of antagonism-mutualistic continuums, the large beak of macaws may have evolved to crack the nuts of palms and also allowing them to transport their large fruits in flight. Our study showed that these macaws were even able to simultaneously transport two large

fruits of bush layer palms, as well as bunches with several fruits of *S. coronata*. HM also uses small pieces of wood and rough leaves as tools to prevent the nuts from slipping during cracking [70,71]. The use of tools to open strong palm nuts, as also observed in monkeys [72], suggests an ongoing evolutionary process, involving palm defense by limiting or precluding predation of the hardest or largest nuts (authors' unpublished data) and the behavioral response of macaws at the limit of their morphological adaptation capabilities, beak power, and shape in this case. On the other hand, the high content of proteins and especially lipids in palm seeds [73,74] can be an adaptation to invest in seed reserves for germination and sapling growth by saving nutrients from the scarce pulp, which is generally discarded by *Anodorhynchus* macaws [71]. The high seed content of specific nutrients can require physiological responses by macaws to support somatic maintenance and reproduction on the basis of a diet dominated by lipids and proteins [75].

The potential mechanisms of palms to manipulate macaws as their seed dispersers, and the macaws' response to cope with them, require further research. Regardless of whether these adaptations may or may not involve co-evolutionary processes, the palm kernels containing the seeds exploited by macaws are typically so large and hard that extant vertebrates have difficulty ingesting and defecating them, especially those of the bush layer palms. Livestock has been proposed as substitutes of extinct megafauna in dispersing the oversized so-called "megafaunal fruits" [15,16]. Livestock, especially cattle, seem to regurgitate rather than defecate seeds of several palm species, although it remains generally unknown if the seeds of the same and different large-fruited plant species are cleaned by consuming the pulp in the mouth or regurgitated after rumination, which can have important implications on seed dispersal distances. Yamashita [27] suggested coevolution between several species of palms and *Anodorhynchus* macaws mediated by the extinct Pleistocene megafauna, inspired by observations of macaws actively looking for piles of seeds regurgitated by cattle. Although this author noted that the nuts accumulated under perching sites where macaws handled them for consumption, he did not consider the possibility that a good fraction of seeds could drop undamaged and germinate. Therefore, a key palm-mammal-macaw mutualistic interaction has thus far been overlooked. However, implicit in the "megafaunal fruit" hypothesis is that seeds should be mainly defecated attending to the attributed role of extinct megafauna [59], paying less attention to the fact that many of the seeds of these fruits do not require gut passage to be efficiently dispersed and recruited. In fact, many large fruits attributed to the megafauna syndrome are legitimately dispersed through stomatochory by macaws and other parrots [17,54] and scatter-hoarding rodents [58].

As hypothesized above, the palm-parrot relationships described here could be viewed under the prism of coevolution. The geographic dispersion and diversification of palms occurred since the late Cretaceous [76,77], whereas fossil parrots are not well known earlier than the Oligocene or Eocene, as a likely consequence of gaps in the fossil record [78]. In fact, multiple molecular studies coincide in delaying parrot origin and radiation to much more ancient periods around the late Cretaceous-Paleogene and onwards, depending on the clade [79–81]. An important diversification of palms took place at this time [76], which might have influenced fruit traits as a consequence of coevolution with parrots. Irrespective of whether this hypothesis could be tested in future studies, parrots could also have played a role in seed dispersal at a time when the megafauna still existed during the Pliocene and Pleistocene. In addition, extinct megafauna was generally younger in evolutionary terms than the large-seeded plants they presumably dispersed in the past, and comparatively much younger than parrots. In fact, the peak of seed size in plants (viewed for floras as a whole) occurred in the early Eocene [82], long before the mammalian megafauna evolved. This historical overview, together with growing evidence on the role of parrots as efficient seed dispersers of oversized fruits, challenges the megafauna syndrome as an anachronism associated with large mammals extinct in the Pleistocene.

### 4.4. The Loss of Palm-Macaw Mutualisms

As is the case for properly measuring biodiversity in terms of species richness [83], more research is needed on seed dispersal in understudied systems [84] for better integrating ecological functions into biodiversity conservation and species recovery planning [6]. The HM and LM are globally threatened species, classified as vulnerable and endangered, respectively, in the IUCN Red List [33,34]. Not only have population sizes decreased drastically in recent decades but so have their distribution ranges. The HM, with a global population estimated at 6500 individuals, is now restricted to three regions of Pantanal, Cerrado, and Amazonia with probably low or even null genetic flow among them [85]. The range contraction of Lear's macaw is even more striking; from an original potential distribution through the Caatinga biome, which covers ca. 845,000 km$^2$ [36], the world's population, last estimated at ca. 1200 individuals, is concentrated in a few breeding sites within a radius of ca. 50 km [86]. The Pantanal, and especially the Cerrado biome, are experiencing a rapid loss of vegetation cover due to deforestation for agriculture and livestock grazing [87,88]. Caatinga has been reduced in the area but mostly strongly degraded by livestock overgrazing [89,90], thus affecting the regeneration of licuri palm (*S. coronata*), whose distribution is restricted to the Caatinga biome [35]. The licuri palm is considered the main food resource for Lear's macaws, so conservation actions aimed at reversing the poor conservation status of this macaw focus on the regeneration and conservation of licuri palm stands [91,92]. Our results challenged previous views on the negative effects of these macaws as exclusive seed predators [25,91], to their key role as legitimate long-distance seed dispersers on the regeneration of palm stands and gene flow among them. Given the large-scale range contraction of both macaw species, key mutualistic plant-macaw interactions have surely disappeared over large surfaces of the studied Neotropical biomes, as may be the case of other plant-parrot mutualistic interactions [32,93]. Moreover, a congeneric species, also presumably specialized in the consumption of large-fruited palms, the glaucous macaw *Anodorhynchus glaucous* [29], was extinct in recent decades, and its range distribution did not overlap with those of the two extant species. As the impacts of Anthropocene defaunation on seed dispersal services are recently gaining much attention (e.g., [94]), further research should focus on the potential impact of the loss of palm-macaw mutualisms. The defaunation of macaws could at least partially explain the current high levels of intrapopulation spatial genetic structure and endogamy in some of our studied palm species [95]. The local and global extinction of these macaws offers the possibility of comparing the spatial and genetic arrangement of palm seedlings and adults in several areas with and without macaw presence to further understand the disruption of dispersal processes. Recovery projects are needed to reverse the high range contraction and population declines of both LM and HM but also to restore their ecological functions (seed dispersal and also food-wasting [65]) at larger spatial scales. There is, however, only one reintroduction project for LM [96], which could be used for assessing the recovery of ecological functions as it has been recently done after the reintroduction of a monkey species [97].

**Author Contributions:** Conceptualization, F.H., G.B., and J.L.T.; methodology, F.H., G.B., and J.L.T.; software, F.V.D.; validation, F.H., G.B., and J.L.T.; formal analysis, J.L.T. and F.V.D.; investigation, J.L.T., F.H., E.P., J.A.D.-L., F.M.F., and G.B.; resources, J.L.T. and N.G.; data curation, F.H. and J.L.T.; writing—original draft preparation, J.L.T. and G.B.; writing—review and editing, F.V.D., G.B., and J.L.T.; visualization, J.L.T.; supervision, J.L.T.; project administration, J.L.T. and E.P.; funding acquisition, J.L.T., E.P., and N.G. All authors have read and agreed to the published version of the manuscript.

**Funding:** This research was funded by Loro Parque Fundacion, grant numbers 101-2015-1 and PP-146-2018-1, and by World Parrot Trust through a long-term (2010–2018) support to Lear's macaw population monitoring (https://www.parrots.org/projects/lears-macaw).

**Acknowledgments:** We thank T. Filadelfo, D. Alves, J. C. Nogueira, M. Cardoso, M. F. Lacerda, R. Alves da Cunha, F. Riera, A.E.A. Barbosa, T. Valença, L. Lima, M. de la Riva, A. Montesinos-Navarro, J. M. Rosa, C. Albano, and P. Lima for their fieldwork assistance. We thank Fundaçao Toyota and Fazenda Caiman, Brazil, for logistic support, Centro de Biodiversidad y Genética (UMSS) for supporting the work as Institución Científica Acreditada, Dirección General de Biodiversidad (MMyMA), Bolivia, for the permits, and Doñana ICTS-RBD for logistic and technical support for fieldwork. S. Young and H. Galal revised the English, and C. Albano, S. Brookes, and M. de la Riva gently provided pictures to illustrate the figures.

**Conflicts of Interest:** The authors declare no conflict of interest. The funders had no role in the design of the study; in the collection, analyses, or interpretation of data; in the writing of the manuscript, or in the decision to publish the results.

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
