# Peer review of "Conserving the Diversity of Ecological Interactions: The Role of Two Threatened Macaw Species as Legitimate Dispersers of “Megafaunal” Fruits"

_diversity, doi:10.3390/d12020045_

Round 1

Reviewer 1 Report

In this paper, Tella and colleagues provide an interesting analysis of the previously overlooked importance of two macaw species across a range of South American biomes as long-distance dispersers of the large seeds of several palm species. The study was able to confirm realized dispersal of large-fruited palms by Anodorhynchus macaws, clearly adds to our knowledge about palm-frugivore interactions, appears methodologically sound, and the conclusions drawn are justified based on the results presented. Overall, the paper is very well written, and the key messages are well articulated. The images make a good addition to the paper, highlighting nicely the various stages in the dispersal process. Overall, I have no major concerns about the paper and merely highlight a few very minor issues below for the authors’ perusal.

l. 68: delete the word “for”

l. 78: frugivores instead of frugivorous

l. 84: “in an Amazonian ecosystem” – can this be changed to “in the Amazonian rainforest”?

l. 122: I would remove fieldwork dates from the subheading 2.1

l. 146: criterion

l. 154: How were you able to distinguish that the macaws were damaging the seeds or not? Were you close enough to be able to actually see that?

l. 169: Dispersal distances – I presume you were not able to infer dispersal distances for the Attalea species in Cerrado since these were only monitored via camera trapping. Please clarify this in the text.

l. 207: please explain abbreviation MLE at first mention

l. 230: genus

Table 2: please italicize all species names

l. 297: could not be obtained

l. 312: dispersal events instead of dispersals

l. 313: unclear what is meant by “mean restriction”, please rephrase

Table 3: “Plant” column – dispersion is not the same as dispersal, please change “all dispersions” to “Total” or similar. Also, in “mean” and “SE” columns, please use decimal points instead of commas (and change elsewhere as applicable throughout the manuscript)

Table 4: legend should go above the table and please italicize all species names

l. 344: AnodorHynchus – typo

l. 355: antagonists

Reviewer 2 Report

This manuscript quantifies palm fruit removal and seed dispersal of plants, mostly palms, by two species of macaw in four biomes of Brazil and Bolivia. The type of data provided are difficult to collect, yet critically important in assessing the ecological value of macaws to palms and of palms to macaws.  The manuscript is well organized and generally clear in presentation.  Methods are reasonable for this type of study, sample sizes are adequate (impressive in some cases!) and statistical tests appropriate.  I question some of the conclusions drawn by the authors (see below) but suspect that in many cases any disagreement can be reconciled by more cautious wording and/or clarification about assumptions.

Most important, the authors make a strong case for seed dispersal services provided by Lear's and Hyacinth macaws. This will come as a surprise to many ornithologists, given the very large and strong bills of Anodorhynchus macaws.  The authors are careful to explain that these birds simultaneously play the roles of seed predator and seed disperser and that even uncommon events of seed dispersal can be demographically important to plants, especially when the dispersal events are long-distance, which is the case here.  I like the way in which the study is placed in the contexts of megafaunal extinction, co-evolution of Anodorhynchus macaws and palms, and conservation.  The Discussion section is a bit long, but worth the space required to adequately address those topics.

I have suggestions to improve the manuscript, which is already strong.

Line 20: I suggest: "...considered a major component..."

Line 43: Likewise, "....there is increasing concern about the loss...."

Line 99.  Note that this statement of purpose and the entire abstract are about palms.  Thus, I was confused when data on other fruits and seeds were presented later in the manuscript.  Those species seemed to appear in the study out of nowhere -- added for no apparent reason.  I'm quite sure that the reason is the authors had valuable data on non-palm species and simply wanted to publish that data.  That's a fine reason, in my opinion.  However, I urge the authors to be more upfront about those data in the Abstract and Introduction (especially lines 98-120).

Line 107.  I question whether dispersal of seeds regurgitated by livestock constitutes tertiary dispersal.  The livestock likely provide primary dispersal by consuming fruits under the parent plant, in which case the macaws would provide secondary dispersal.  Please clarify.

Line 140: "We found only one case...."

Lines 141-143: I don't understand: "we identified...of both species". What are the two (?) species of palm and how was their identification resolved.

Line 148: "This inconsistency..."

Lines 156-168: When I read this paragraph, I assumed the study was exclusively about four species of palm, two in the Bolivian pantanal and two in the Cerrado.  A few pages later, it became clear that six species of palm and many species of non-palm were included in the study.  This mismatch between what's summarized in the Abstract, framed in the Introduction, and described in Methods vs. presented in Results is confusing.  See prior comment for a suggested fix.

Line 165.  More information about the camera traps is needed.  How often did they take photos, once activated by a macaw?  Did they record videos?  Most important: how was removal attributed to a given bird if a complete sequence of images for a visit was not available?

Lines 184-191.  Here is where I think it's important to be careful about assumptions and methodological limitations. Specifically, the authors should state that they assume seeds found near or under perches where they observed macaws were placed there by macaws.  I don't doubt that most of them were but certainly there are other potential dispersers that use exposed perches (e.g., cotingas -- PNAS 1998, https://doi.org/10.1073/pnas.95.11.6204 ) and terrestrial frugivores that could scatter seeds over a very large area.  Similarly, the authors should acknowledge that their sampling under perches is a snapshot approach that prevents any understanding of seed and seedling dynamics; many of the seeds (especially the viable ones?) are likely to be removed by terrestrial rodents and many of the seedlings are unlikely to survive for long.  How do such processes affect the authors' interpretation?  Finally, the authors have no metric of seed viability -- they assume that intact seeds are viable, but why would macaws discard "good" (viable) seeds?  I doubt it's because they are sloppy eaters.  Could those seeds be somehow infested or otherwise damaged in a non-apparent way?

Line 223. "work or invasive"

Line 230: I don't understand what "...the Genus..." means.

Lines 248-249: "...by carrying in the bill but were sometimes carried with the feet..."

Table 3.  I suggest "all dispersal events" instead of "all dispersions"

Line 318.  Why is it "surprising"?

Line 328. It should be clearly stated that "the capacity to germinate" is an assumption.  The authors provide no data on seed viability, only the observation of seedlings, which indicate that some seeds were viable.

Line 460. "...able to simultaneously transport..."

Line 477. "...the oversized so-called..."
